# Deep reinforcement learning for active flow control in a turbulent separation bubble

Bernat Font [1,2] ✉, Francisco Alcántara-Ávila [3], Jean Rabault [4], Ricardo Vinuesa [3] ✉ & Oriol Lehmkuhl [2]

The control efficacy of deep reinforcement learning (DRL) compared with classical periodic forcing is numerically assessed for a turbulent separation bubble (TSB). We show that a control strategy learned on a coarse grid works on a fine grid as long as the coarse grid captures main flow features. This allows to significantly reduce the computational cost of DRL training in a turbulent-flow environment. On the fine grid, the periodic control is able to reduce the TSB area by 6.8%, while the DRL-based control achieves 9.0% reduction. Furthermore, the DRL agent provides a smoother control strategy while conserving momentum instantaneously. The physical analysis of the DRL control strategy reveals the production of large-scale counter-rotating vortices by adjacent actuator pairs. It is shown that the DRL agent acts on a wide range of frequencies to sustain these vortices in time. Last, we also introduce our computational fluid dynamics and DRL open-source framework suited for the next generation of exascale computing machines.

Turbulence plays a key role in a wide range of applications, as turbulent flows are present in almost all industrial processes. While significant advances have been made in the theoretical aspects of turbulence during the past half century, there are still numerous open questions regarding these flows. Evidence of the complexity of turbulence is the fact that the proof of a unique and smooth solution of the Navier–Stokes equations, which govern the behavior of turbulent flows, was selected as one of the Millennium Prize Problems by the Clay Mathematics Institute in 2000[1]. Turbulence has practical implications for the energy costs of the transport industry. For example, with an increasing number of flights, the aviation sector has a major impact on total $CO_2$ emissions. According to Owen et al.[2], aviation is responsible for 12% of the total $CO_2$ emissions in the transport sector. A significant fraction of the energy consumption associated with aviation is used to overcome turbulent drag. Similar conclusions can be drawn for other types of vehicles from automotive to maritime transport. Therefore, even a small improvement in the aerodynamic efficiency of these vehicles can have a great impact on the reduction of global $CO_2$ emissions.

In the case of aircrafts, significant aerodynamic losses arise when the flow detaches from the wing surface, leading to an increase in drag and fuel consumption. This phenomenon occurs especially in critical situations such as takeoff, landing, or due to large-scale turbulence in the free-stream flow during cruise conditions. Being able to optimize the flow around aerodynamic surfaces can help to reduce drag and increase aerodynamic efficiency. Flow control (FC) aims at finding adequate strategies to modify certain flow properties, for example, enhancing/reducing turbulence or heat transfer, reducing the aerodynamic drag or improving the maneuverability of vehicles. Traditionally there are two different branches in the FC field: passive flow control (PFC) and active flow control (AFC). While PFC usually focuses on geometry modifications, AFC allows a closed-loop interaction with the flow. Returning to the aircraft example, the wing suction side exhibits an adverse pressure gradient (APG) that can induce flow separation at high angles of attack. In such cases, the flow can separate and then reattach to the surface creating a turbulent separation bubble (TSB). AFC can have a great impact on the wing aerodynamic performance under these circumstances by employing a proper control strategy which can reduce the TSB area. There are countless applications of FC documented in the literature, and the reader is referred to refs. 3–5 for comprehensive reviews.

[1]Faculty of Mechanical Engineering, Delft University of Technology, Delft, Netherlands. [2]Barcelona Supercomputing Center, Barcelona, Spain. [3]FLOW, Engineering Mechanics, KTH Royal Institute of Technology, Stockholm, Sweden. [4]Independent researcher, Oslo, Norway. ✉e-mail: b.font@tudelft.nl; rvinuesa@mech.kth.se

Using computation fluid dynamics (CFD) to study theoretical aspects of FC in a wing with a TSB is computationally challenging. A first approach to the problem is the use of a turbulent boundary layer (TBL) subjected to an APG that generates the TSB. This canonical domain resembles the suction side of the wing in a simplified manner. TBLs under an APG are characterized by a wider parametric space compared to zero-pressure gradient (ZPG) TBLs. According to the parametric study performed by Monty et al.[6], the most influential parameters are: a) the Rotta–Clauser pressure-gradient parameter $\beta = (\delta^*/\tau_w)\mathrm{d}_x P$, where $\tau_w$ is the wall-shear stress, $P$ is the static pressure, $x$ is the streamwise direction coordinate, and $\delta^*$ is the displacement thickness. This parameter represents the pressure-to-viscous force ratio. b) The friction Reynolds number $Re_\tau = \delta^* u_\tau/\nu$, where $u_\tau = \sqrt{\tau_w/\rho}$ is the friction velocity, $\rho$ is the fluid density, and $\nu$ is the kinematic viscosity. This parameter represents the ratio between the large and the small scales of the flow, i.e., how turbulent the flow is. c) the acceleration parameter $K = (\nu/u_\infty^2)\mathrm{d}_x u_\infty$, where $u_\infty$ is the local free-stream velocity. This parameter represents the equilibrium state of the boundary layer. Several studies in the literature have investigated TBLs exposed to an APG under various parametric conditions, employing both numerical and experimental approaches. These examples include direct numerical simulations (DNS)[7–11], well-resolved large-eddy simulation (LES)[12–14], and applying control in actual experiments[6,15,16].

AFC has been the focus of numerous TSB studies for both canonical and practical flows. The latter refers to wing-like configurations, where the surface curvature and/or angle of attack induce flow separation. Although it might be possible to prevent flow separation rather than reducing it, this is not always feasible. Therefore, numerous attempts and approaches have been used to tackle the reduction of flow separation as reflected in the extensive literature on this topic. Some examples of the different methods used are the injection of high streamwise momentum near the wall[3,17], the use of near-wall flow disturbers[18], the use of plasma actuators[19], periodic excitation[20], vortex generators[21], or the use of blowing and suction resembling synthetic jets[22,23], among others. The latter is the method used in this work. A typical strategy when applying blowing and suction is to set the wall-normal velocity of the control jets to a sinusoidal function determined by the control signal amplitude and frequency. In this way, the resulting mass flow through the jets is zero after an actuation period ends. This is known as the zero-net-mass-flux (ZNMF) periodic control. You and Moin[24] applied ZNMF periodic forcing by connecting the pressure and suction sides of a NACA 0015 airfoil. Using harmonic blowing and suction, they found that this control effectively delayed the separation of the boundary layer at high angles of attack. Atzori et al.[25] focused on uniform blowing and uniform suction applied to the suction side of a NACA 4412 airfoil, finding that AFC had a larger impact on the APG TBL compared to ZPG TBL. Lehmkuhl et al.[26] used LES to investigate periodic forcing applied on the suction side of an SD7003 airfoil wing and the JAXA standard model. They found that the periodic forcing successfully eliminates the recirculation bubble in both cases. The parameter that describes the frequency of the actuation is the Strouhal number, $St = fL_b/U_\infty$, where $f$ is the actuation frequency, $L_b$ is, in this context, the length of the uncontrolled TSB and $U_\infty$ is the free-streamwise velocity. While most of the works have demonstrated that $St \sim \mathcal{O}(1)$ usually provides an effective control[23,27], different conclusions are drawn when actuating at higher values of $St$ depending on the problem[28–30]. On the other hand, applying periodic blowing and suction at lower frequencies has been demonstrated to be an efficient control strategy for TBL separation control[31,32].

For canonical flows, i.e., TBL, there are different possibilities to generate a TSB. A common approach in CFD is to set a vertical velocity at the top boundary of the domain to create the APG. Two different options are usually employed: suction-blowing (SB) or suction-only (SO). The SB approach has the advantage of instantaneously conserving mass flux across the domain boundaries and yields a more stable recirculation bubble caused by the favorable pressure gradient (FPG)

that follows the APG. For the SO case, the flow is naturally reattached by the turbulent diffusion of momentum, so it can better resemble the separation type found in wings[32]. Cho et al.[31] focused on a TSB generated by SB on the top boundary of the domain. Using periodic forcing as described above, and sweeping across a range of characteristic frequencies $S_t$, they found that low frequencies $St < 0.5$ tend to reduce the separation bubble and high frequencies $St > 1.56$ induce an intermittent separation/reattachment pattern. Wu et al.[32] investigated a TSB generated by SO and the effect of periodic forcing for a range of $St$. It was found that the frequency $St = 0.45$ as well as a higher frequency $St = 1.125$ yield a 50% reduction of the TSB. On the other hand, a much higher frequency of $St = 4.5$ did not affect the TSB.

Over the last decade, following the increase of computational power available, artificial intelligence has emerged as a substitute of classical methods to solve different problems, as reported in ref. 33. There are numerous examples where machine learning (ML) has been successfully applied to solve experimental and numerical closed-loop FC problems[5]. Some of these cases are based on genetic programming for broad-band frequency turbulence[34], or the method employed in this study, i.e., reinforcement learning (RL). RL stands out as a subset of ML that operates without the requirement of a-priori data for the training of a neural network (NN). When employing a deep neural network (DNN) in RL, the approach is referred to as deep reinforcement learning (DRL). DRL has proven to be a very effective approach for developing AFC strategies in many scenarios where the training data is generated on the fly while the control is happening. Therefore, one of the main goals of this study is to compare the performance of the classical periodic control with that obtained by DRL and investigate how ML techniques can improve upon classical control theory. The main motivation behind the use of DRL is to allow the controlling agent freedom in selecting the most appropriate action for a given flow state, instead of using a single frequency as in classical periodic control. Therefore, DRL allows for an unconstrained closed-loop control strategy that can adapt to the system dynamics based on previously learned experiences. The typical DRL setup consists of two main entities: an agent, which is composed by DNNs and accompanying optimization algorithms, and an environment, which in this case corresponds to the flow simulator. These two entities interact through three communication channels: a) the state $s(t)$, which is sent from the environment to the agent. The state can be a total or partial observation of the environment at a given time $t$. In the case of a numerical simulation, $s$ is usually the value of a set of probes distributed in the domain regions of interest, e.g., velocity or pressure around and in the TSB. b) the action $a(t)$, which is sent from the agent to the environment. The action is a modification to the environment, e.g., a new boundary condition which is related with the control strategy. For example, the action can be the mass flow rate of synthetic jet actuators, a modulation of the temperature profile in a wall, among others. c) the reward $r(t)$, which is sent from the environment to the agent. The reward is the parameter that represents the environment fitness with respect to an optimization goal, e.g., the drag coefficient, or the size of the TSB. Therefore, the agent will try to optimize a reward $r$ through the use of an action $a$ which is selected based on a state $s$, i.e., the agent will use a policy $\pi(a|s)$ to optimize the reward. The policy defines a probabilistic mapping from the current state $s$ to the action $a$. During training, the agent works in the so-called exploration mode by adding noise to the action sampling process. In this way, new dynamics can be explored and learned by the algorithm. During a training episode, a collection of $(s, r, a)$ triplets is generated, also known as a trajectory. After an episode is finalized, the resulting trajectory is used to update the NN weights so that the accumulated reward expectation is maximized. Once the training is finalized, the agent is used in the so-called deterministic mode, where the most probable actions are selected, i.e., the maximum of the policy distribution $\pi(a|s)$. A general overview on the choice of policy is given in the "Methods" section.

In recent years, the application of DRL to AFC problems has grown exponentially as shown in the increasing amount of literature on the topic. A wide overview of the most relevant works is presented by Vignon et al.[35], where the main advances, tendencies, and types of problems addressed by DRL for AFC are discussed. Some of the most characteristic cases studied in the literature include drag reduction of a cylinder, both in two dimensions (2D)[36–41] and three dimensions (3D)[42–44], noting that Fan et al.[43] and Amico et al.[44] showed successful DRL-based control for highly turbulent flows in experiments; convective heat reduction in Rayleigh-Bénard convection problems[45,46]; reduction of the skin-friction coefficient in turbulent channels[47,48]; and turbulence modeling[49–51]. Currently, the community is working towards expanding the use of DRL to higher complexity and more realistic cases. We note that the implementation of DRL-defined control functions for real-world applications is more challenging than traditional periodic forcing. However, new experimental frameworks such as Dong et al.[52] are now emerging. For realistic applications, where the non-ideal transfer function of the physical system can introduce delays and inertia in the actual control values, we expect that the DRL agent will be able, with proper learning against the full system including imperfect actuators, to anticipate these additional challenges. Indeed, the actuators' transfer function will then be part, from the DRL agent viewpoint, of the transfer function of the whole system to control. As a consequence, the DRL agent can learn to compensate for the imperfect behavior of real-world actuators through online training in the laboratory, as demonstrated in studies such as Fan et al.[43] and Dong et al.[52].

From the computational standpoint, several novel DRL techniques that can accelerate the training process of a control strategy have been adopted. A first approach is to simulate multiple environments in parallel, also known as multi-environment DRL. With this approach, the agent trains faster by generating multiple experiences in parallel. This method has an almost perfect scaling as shown by Rabault and Kuhnle[53]. A second approach, orthogonal to the multi-environment DRL, is to use a multi-agent reinforcement learning (MARL) method. First introduced by Belus et al.[54] and later named by Vignon et al.[46], the MARL method exploits the domain spatial invariants so that the dimensionality of the actuation space is reduced. For example, consider a set of multiple synthetic jets placed along the span of a circular cylinder. Since the flow is statistically invariant along the spanwise direction, every individual jet can be treated separately within its own span subdomain (or MARL pseudo-environment). In this approach, rather than predicting multiple actions simultaneously, individual agents are dedicated to each MARL pseudo-environment, each responsible for predicting a single action. The key is that all these agents share the same policy so that every individual learning experience is shared among all the agents. The advantage of this approach is the reduction in the number of combinations that yield an overall positive action, hence avoiding the curse of dimensionality. As reported in Suárez et al.[42] and Vignon et al.[46], using MARL allows the agent to effectively learn the system dynamics and provide a positive control strategy even when multiple actuators are used. Defining the number of parallel environments as $N_e$ and the number of parallel MARL pseudo-environments within each environment as $N_{pe}$, a total of $N_e N_{pe}$ trajectories can be sampled in parallel, and the training time of an agent can be greatly reduced. Last, the computational cost of DRL training can be reduced by performing transfer learning, as demonstrated in refs. 41,55, where training a DRL agent at a low-Reynolds-number flow can still yield successful control at (more computationally expensive) higher-Reynolds-number flows.

From a CFD perspective, there has been increasing interest in utilizing graphics processing units (GPUs) for conducting computationally intensive simulations. In this context, we developed the SmartSOD2D framework, originally a fork of RELEXI[51,56]. This framework enables CFD simulations, which constitute the primary computational workload, to be executed on GPUs, while the DRL algorithm, which has a lower computational cost, runs on the central processing unit (CPU). The high-order spectral-element-method solver SOD2D[57] is used to run the CFD simulations on the GPUs, and the TF-agents libraries[58] are used for the DRL model part in the CPU. Furthermore, this framework uses the SmartSim[59] library to allow communications between the agent and the environment through an in-memory database, reducing the writing/reading time into/from disk compared to, e.g., ref. 42. Owing to the scalability of the CFD simulation on GPUs, SmartSOD2D is suited for the next generation of exascale computing machines. We make the whole SmartSOD2D framework open source together with the present work.

## Results
### Periodic control
The results using the classical periodic forcing are presented next, and serve as reference for the DRL control. First, a non-actuated simulation, defined as the baseline, is run for 5000 convective time units (CTU), corresponding to the simulation time normalized by $\delta_0^*$, the boundary layer displacement thickness at the inlet, and $U_\infty$, the free-stream velocity. We note that $\delta_0^*$ and $U_\infty$ are used to respectively normalize length and velocity (and hence time) quantities throughout this work. This baseline is run until the flow is statistically stationary and serves as an initialization for the open-loop periodic control. Then, statistics are recorded during 15000 CTU for both a coarse and a fine grid applying the aforementioned periodic control. While the main objective is to successfully control the fine-grid case, which captures a wider range of scales thus being more representative of the physical system, training the DRL agent on this grid is computationally prohibitive. Hence, we train the agent on a coarse grid which still captures the main physics of the problem and significantly reduces the computational cost of training, as previously used in e.g., refs. 41,47,55,60. Afterwards, the model can be trained further on the fine grid via transfer learning, i.e., using the optimized NN weights already available as the training starting point on the fine grid. Or, alternatively, given that similar dynamics are captured in both coarse and fine grids, the controlling agent can be directly applied to the fine-grid simulation without further training.

Figure 1 shows in black shade the time-averaged recirculation region, defined as the region where $\overline{u}<0$ on the $xz$-plane of the first off-wall node ($y_1$) for the fine ($y_1 = 0.0635$) and coarse ($y_1 = 0.0864$) grids, respectively. The non-actuated fine-grid case (top left) exhibits a recirculation bubble generated by the SB top-boundary condition that qualitatively spans from $x = 225$ to $x = 350$. On the other hand, the bubble undergoing periodic control (top middle) qualitatively spans a smaller region of the domain and new separation bubbles are visible near the actuators. The reduction of the TSB in the downstream region under periodic control is a phenomenon also observed by Cho et al.[31] in a similar SB APG TBL setup. The bubble area can be better quantified by the time-averaged characteristic recirculation length $\overline{l_x}$, which estimates the normalized recirculation area and is formally defined in the "methods" section. As presented in Table 1, a reduction of 6.8% in $\overline{l_x}$ is obtained for the fine grid when using the periodic control. We also note that the normalized standard deviation of $\overline{l_x}$, i.e., $\sigma(\overline{l_x})$, is similar to the root-mean-square (RMS) value of the forcing signal, i.e., $\sigma(\overline{l_x})/\overline{l_x} \sim A_{ac}/\sqrt{2}$. This implies that the periodic-forcing-control effect dominates the TSB dynamics. This phenomenon is visible in Fig. 2, where we show the temporal signals of $l_x$ on the non-actuated and controlled cases. Since the wall-normal velocity profile imposed at the top of the domain generates a FPG during the blowing section of the SB setup, the reattachment point of the TSB has less freedom compared to a SO setup. In the latter case, a reduction of the TSB up to 50% can be observed when using periodic forcing under the correct actuation frequency, as reported by Wu et al.[32].

The instantaneous flow field of the TSB actuated with the periodic control is depicted in Fig. 3b. A large-scale spanwise vortical structure

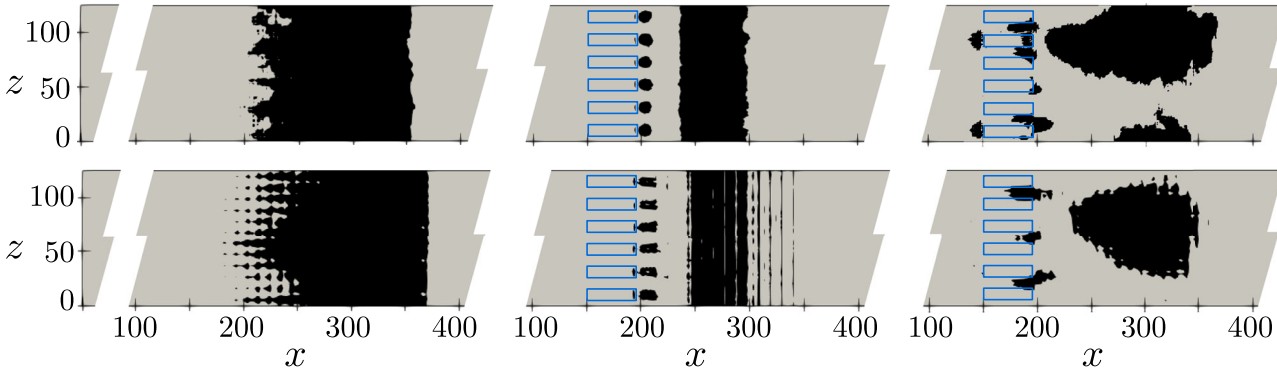

**Fig. 1 | Time-averaged recirculation region ($\overline{u}<0$) at the $xz$-plane of the first off-wall node for the fine grid (top) and coarse grid (bottom).** From left to right: non-actuated, periodic control, DRL control.

is observed as a result of the actuation. The interaction of these structures with the TSB effectively reduces the size of the bubble. Additionally, the power spectra of the streamwise velocity component ($u$) at witness points located inside the TSB and on the reattaching shear layer are displayed in Fig. 4. The actuation is clearly visible through the peak arising precisely at $f_{ac}$ = 0.0025, while a harmonic peak is also present at a higher frequency. The selected periodic-control frequency and amplitude are justified in the "Methods" section.

The main objective of the coarse–fine grid comparison is to assess whether the main flow features and the general physics are preserved so that the coarse grid can be used for training the DRL model. Once the DRL model is obtained, it is then directly tested in the fine grid. Coarse-grid results depicting the time-averaged recirculation region are displayed in Fig. 1 (bottom). In general, we observe a very similar flow structure compared with the fine-grid results. Also, the TSB is slightly larger in the coarse grid case, as quantified in Table 1. The characteristic recirculation length is reduced by 15.7% when applying the periodic control on the coarse grid. Similarly to the fine grid, the standard deviation of this metric is significantly increased as a result of the periodic forcing. Spectra of the coarse-grid simulation are also shown in Fig. 4. While the general trend is in good agreement with the fine-grid results, the spectrum of the probe located on the reattaching shear layer region presents some discrepancies. This can be expected since the stretching of the coarse grid significantly reduces the resolution far from the wall. Nevertheless, the main flow features of the fine-grid simulation are preserved, and we conclude that the coarse grid is adequate for training the DRL model.

## DRL control
The training metrics are presented in the "Methods" section. Once the DRL agent has been trained, we test the learnt control strategy under deterministic behavior, i.e., selecting the best possible action from the

action probability distribution, or in other words, the mean of $\pi(a|s)$. First we focus on the results of applying the agent in the coarse grid. The DRL model is run in deterministic mode for 20000 CTU using a baseline (non-actuated) snapshot as initial condition. Table 1 shows the time-averaged characteristic recirculation length of the periodic control and DRL control cases for the last 15000 CTU and the percentage reduction of the TSB compared with the non-actuated case. The DRL control provides a larger reduction of the TSB compared to the classical periodic forcing, respectively 25.3% and 15.7%. The temporal signal $l_x$ resulting from the DRL-controlled TSB is plotted in Fig. 2 (right), where it can be observed that the DRL control quickly reduces the TSB length. Importantly, the DRL control presents much better stability in time than the periodic control, hence the recirculation bubble evolves smoothly and no sudden oscillations are present. While the periodic control also displays a reduction of the bubble, spurious peaks can be observed in the signal. This can arise from the fact that the DRL control strategy is forced to conserve mass in space, i.e., instantaneously (and therefore in time too), while the periodic forcing is only conserving mass over each period of actuation. The instantaneous conservation of mass in incompressible flows is important not only from the physical point of view, but also numerically in the pressure solver. Since our numerical scheme does not correct for the instantaneous mass imbalance of the periodic forcing, the aforementioned spurious oscillations arise, which are visible by the $l_x$ signal for the periodic control.

The time-averaged recirculation regions for both periodic control and DRL control are shown in Fig. 1. The DRL control yields a different flow-field structure. A strong recirculation region located at $x = 180$, on top of the actuators, is observed. Since DRL actuators are spanwise-paired with equally positive and negative mass flow rates, this can eventually generate streamwise structures that interact with the TBL resulting in these small separation bubbles. The time signals of the actions set by the DRL control agent are shown in Fig. 5. The coarse-grid case shows that actuators oscillate between the maximum and minimum allowed values ($|v_{ac}|_{max} = 0.3$), i.e., the amplitude used in the periodic-control signal, resulting in a bang–bang control, while short transitional phases arise as well. This is consistent with previous results, in example[47], which observed that DRL control tends to include bang-bang-like features when controlling turbulent flows. Note that despite the actuation signal resembling a bang–bang control, the actual values imposed on the control surfaces are smoothed over time, as explained later in the "Methods" section. This results in a more realistic and applicable control signal.

The agent obtained by training on the coarse grid is used in deterministic mode on the fine grid without further training. The DRL control is applied for 20000 CTU and the recirculation bubble characteristic length is averaged over the last 15000 CTU. Table 1 shows that the agent trained in the coarse grid is capable of reducing the

**Table 1 | Time-averaged characteristic length $\overline{l_x}$ and standard deviation for the non-actuated (baseline), periodic-control and DRL-control cases for both fine and coarse grids**

|  | $\overline{l_x}$ | | | $\widetilde{l_x}$ | | $1-\overline{l_x}/\overline{l_x^*}$ | |
| --- | --- | --- | --- | --- | --- | --- | --- |
| Case | Baseline | Periodic | DRL | Periodic | DRL | Periodic | DRL |
| Fine | 143.0 ± 8.6 | 133.3 ± 27.2 | 130.0 ± 9.9 | | | 6.8% | 9.0% |
| Coarse | 153.6 ± 8.3 | 129.5 ± 33.4 | 114.7 ± 6.7 | | | 15.7% | 25.3% |

The reduction of the characteristic recirculation length with respect to the baseline case is also shown. Note that the DRL control on the fine grid is obtained using the agent trained on the coarse grid, and no further training is performed on the fine grid.
The average is computed for 15000 CTU after discarding the initial transient 5000 CTU.

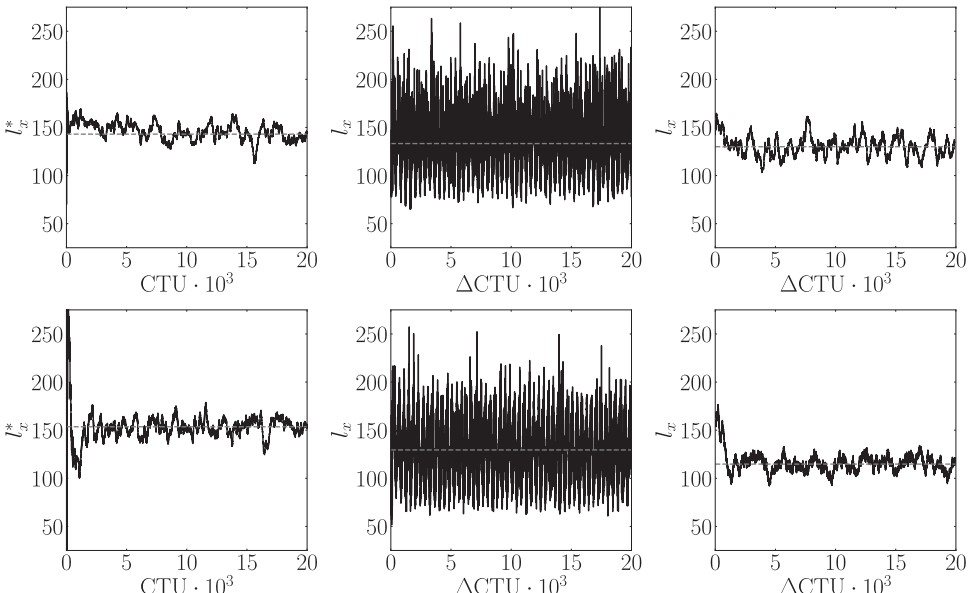

**Fig. 2 | Temporal signal of the characteristic recirculation length the fine-grid cases (top), coarse-grid cases (bottom).** From left to right: baseline, periodic control, DRL control. The dashed gray line indicates the time averaged value as reported in Table 1. Note that the DRL-control on the fine grid is obtained using the agent trained on the coarse grid.

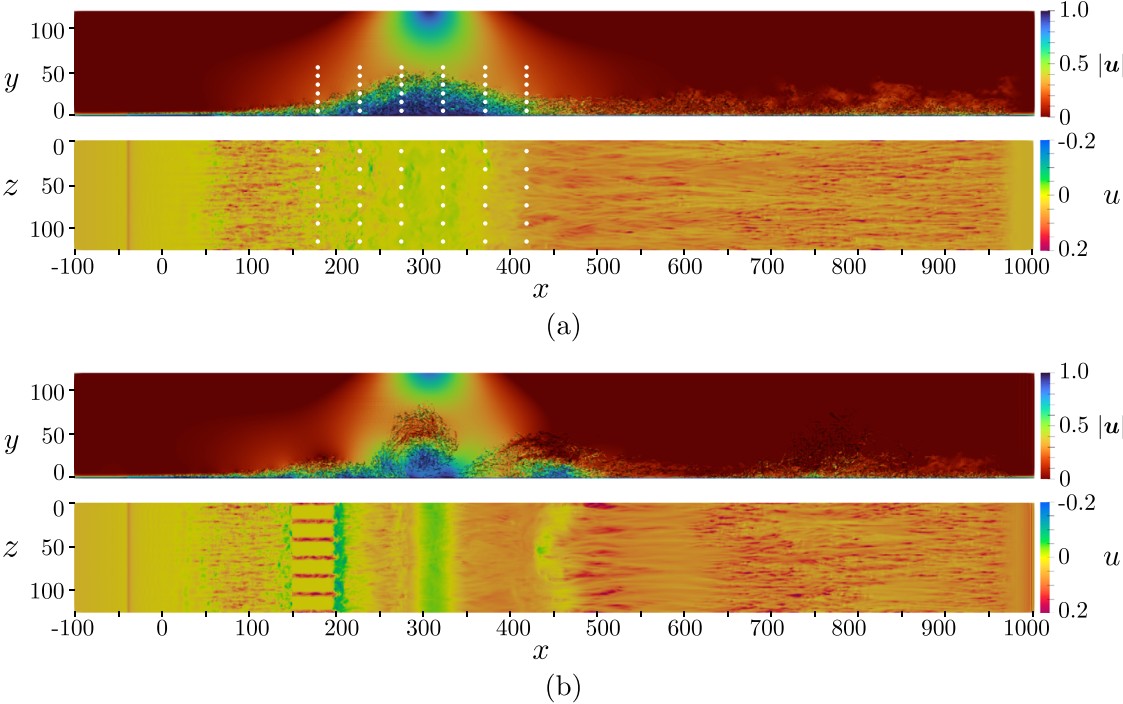

**Fig. 3 | Flow visualization.** Instantaneous snapshot of the fine-grid cases for (**a**) non-actuated, (**b**) periodic control. For each subfigure, top: $xy$-plane at $z = L_z/2$ displaying the vortical structures captured by the Q criterion[82] and colored by velocity magnitude, and bottom: $xz$-plane of the first off-wall node displaying the streamwise $u$ velocity component. For the periodic control, the actuators are found within $x \in [150, 195]$. The locations of the witness points which capture the flow state during the DRL control are also displayed in subfigure (**a**).

bubble of recirculation by 9.0%, improving the 6.8% reduction achieved using the periodic control. Furthermore, from a qualitative point of view, the temporal evolution of $l_x$ is more stable when using the DRL agent than with the periodic control, where there is a clear influence of the phase of the actuation. It is worth noting that the control agent was never trained on the fine grid. Therefore, the control strategy on the fine grid may be possibly improved with transfer learning, i.e., using the agent trained on the coarse grid and train it further in the fine grid. The control strategy found in this case resembles a two-step control, where two actuators are saturated, and the third one oscillates between the two allowed maxima. Unlike the coarse-grid case, when the agent is applied on the fine grid, the MARL pseudo-environment performing the two-step oscillations is always the same. However, despite this difference, the invariance in the

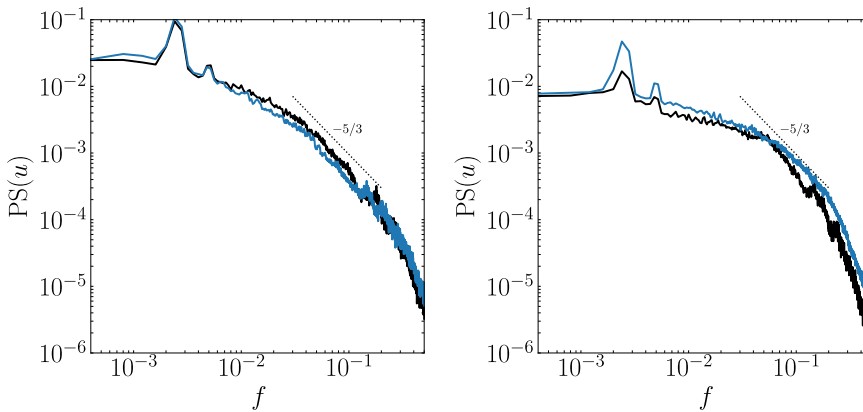

**Fig. 4 | Power spectrum (PS) of the streamwise velocity component $u$ of the periodic-control case sampled at two different locations: $(x, y) = (324, 15)$ (left) and $(x, y) = (372, 55)$ (right) for the fine-grid (blue) and coarse-grid (black) cases.** The PS is computed using the Welch method for a temporal signal of 15000 CTU in total, split into 6 segments with 50% overlap and uses Hann windowing. Additionally, 6 PS are computed along $z$ for each $(x, y)$ location which are then averaged and result in the displayed spectra. The black-dashed line shows a $-5/3$ power law.

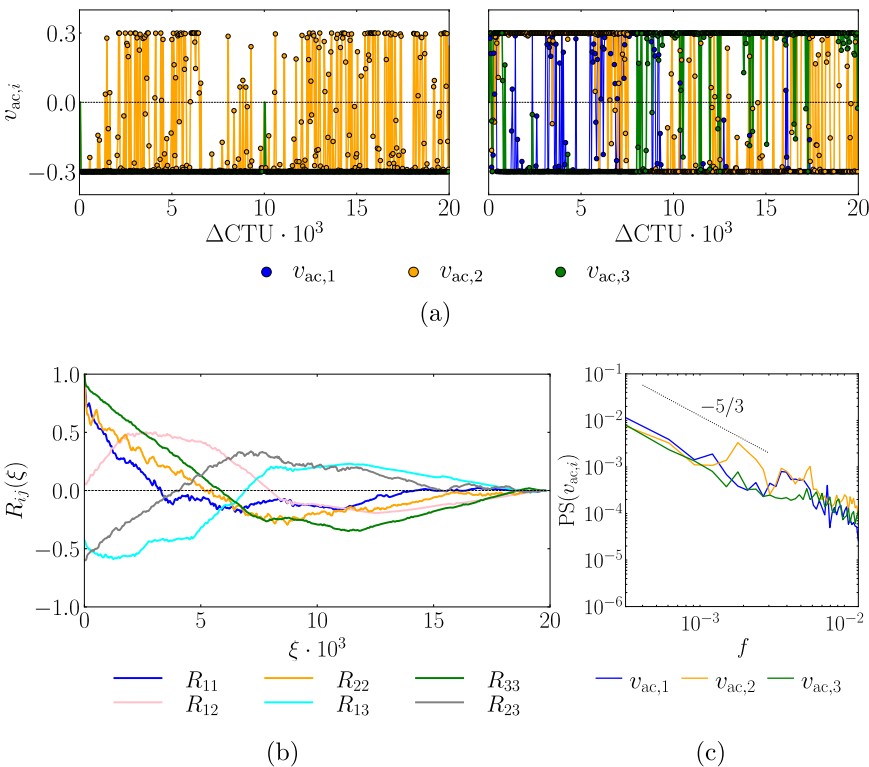

**Fig. 5 | Analysis of actuation signals. a** Temporal signal of the DRL-based actuators for the fine-grid (left) and coarse-grid (right) cases.The exponential smoothing between two discrete actions that sets the instantaneous mass-flow rate of the actuators is also displayed. Note that actuators 1 and 3 overlap on the fine-grid case. **b** Normalized cross-correlation between actuation signals of the coarse-grid case, where $\xi$ is the correlation interval. **c** Power spectrum of the actuation signals of the coarse-grid case computed with the Welch method using 6 splits with 50% overlap and no windowing.

spanwise direction makes both results equivalent. These results show that it is possible to achieve excellent control results on the fine grid by using the agent trained on the coarse one.

The physical interpretation of the DRL control is derived from figures 5b, c and 6. The cross-correlation analysis of the control signals exhibits a large temporal scale, where significant correlation values ($R \gtrsim 0.5$) are found for temporal correlation intervals up to $\xi \simeq 2500$ CTU. This hints that the actuators might be generating a lasting time-sustained structure in the flow. Furthermore, the power spectrum of the control signals shows that a wide frequency range is explored, and a peak is found near the natural TSB breathing

frequency ($f \approx 0.002$) for the $v_{ac,3}$ signal. It can also be noted that the spectral distributions of signal energy decay following a $-5/3$ power law, which is the analytical and experimental rate found in the inertial subrange of turbulence (also noted in Fig. 4). Moreover, the time-averaged flow fields in Fig. 6, which better capture the coherent turbulent flow structures, confirm the presence of streamwise vortices generated by the actuator pairs; these vortices tilted by the top SB condition creating oblique coherent structures. Note that the low-frequency vortices positively interact with the TSB causing its attenuation, while being constantly sustained by the mid- and high-frequency adjustments made by the DRL agent. Interestingly, the use

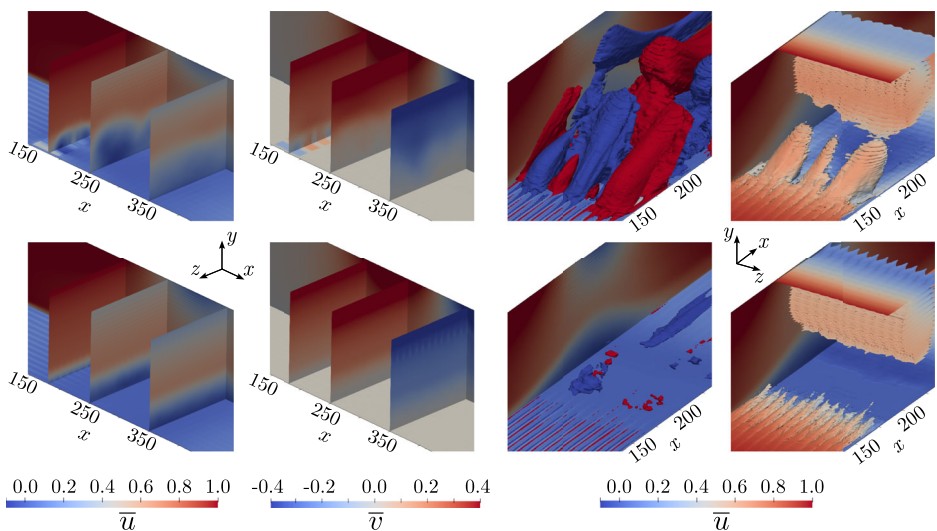

**Fig. 6 | Time-averaged coarse-grid flow fields for the DRL control (top) and uncontrolled (bottom). The x-axis is aligned with the streamwise flow direction.** The first and second columns show the time-averaged streamwise and wall-normal velocity components, respectively. The third column shows the iso-surface of the time-averaged spanwise velocity $|\overline{w}| = 0.02$ (red for positive), while the wall and the periodic plane are colored by $\overline{u}$. The fourth column shows the iso-surface of the time-averaged vorticity magnitude $|\overline{\boldsymbol{\omega}}| = 0.04$, while the wall and the periodic plane are colored by $\overline{u}$.

of streamwise vortices for flow reattachment and flow-separation mitigation is a known control technique that has been discussed and used for many decades (see e.g., refs. [20,61,62] and references therein). This can be achieved either by using micro-vortex generators, as pointed in ref. [61], by using jets[63], or even plasma actuators[64–66]. At a low Reynolds number, DRL control of plasma actuators was successfully applied in ref. [67] to reduce flow-induced forces on a 2D square cylinder. In the present study, we observe that the DRL agent is able to re-discover the large-scale streamwise vortices strategy on its own for this high-Reynolds-number 3D flow, while also learning to adjust the wall-normal jets to sustain the vortices for long time periods and obtain effective flow reattachment.

## Discussion

In the present work, we investigate the efficacy of classical periodic control and deep reinforcement learning (DRL) control in reducing a turbulent separation bubble (TSB) generated by a suction and blowing (SB) boundary condition in a turbulent boundary layer (TBL) flow.

The classical periodic control, relying on harmonic forcing in time, reduces the TSB area by approximately 6.8% and 15.7% on a fine and a coarse grid, respectively. This corresponds well to previous literature of similar configurations. Additionally, a spectral analysis highlights the impact of the actuators on the flow, with clear peaks at the actuation frequency and harmonics. The coarse-grid results demonstrate the preservation of the main flow features, validating the use of the coarse grid for the subsequent training of the DRL model. This classical control configuration was selected based on the recommendations in the literature, e.g.,[32] as described in the methods.

The DRL control demonstrated successful learning of a control strategy that can effectively reduce the TSB area by 25.3% and 9.0% in the coarse and fine grid cases, respectively. Compared with the periodic control, the DRL has the freedom to set the optimal mass flow rate of the actuators for a given environment state, hence constructing a complex control signal that can embed multiple frequencies. The training was performed using 24 parallel MARL pseudo-environments running on 8 GPUs and took 144 hours (6 days) in total, equivalent to 1152 GPU-hours on the coarse grid. The robustness of the DRL method allows performing less expensive training of the agent on the coarse grid with still good performance on the fine grid. The physical analysis of the control strategy shows that each DRL-controlled actuator pair generates a streamwise vortical structure which is sustained for a long time period, resembling classical separation-control techniques based on streamwise vortices such as vortex-generator jets or plasma actuators. Remarkably, the DRL controller is able to re-discover this control strategy independently, and to stabilize the vortices generated by including complex high-frequency features in the control signals. To the best of our knowledge, the current numerical results present a successful application of DRL-based control at one of the highest Reynolds numbers to this date in a completely turbulent flow.

In order to make this work possible, we have developed a new open-source framework, SmartSOD2D, that couples DRL libraries and the CFD solver. More specifically, SmartSOD2D integrates the SOD2D CFD solver (a novel multi-GPU spectral element solver developed at Barcelona Supercomputing Center) with the DRL model via SmartSim, which allows a fast communication between the agent and the environment through memory. Up to the knowledge of the authors, SmartSOD2D is the first framework in DRL for AFC that fully leverages GPUs for simulating the CFD environments. Since the CFD simulation, where most of the computational cost lies, is conducted in the GPUs, SmartSOD2D has an excellent scalability suited for the next generation of exascale computing machines, opening the possibility of using DRL for AFC for high-Reynolds-number flows and complex geometries. Also, SmartSOD2D incorporates the MARL approach which is key for problems with spatial invariance and to obtain successful learning on distributed-input distributed-output systems. Links to the different components of the open-source CFD-DRL framework are given in the "Methods" section.

## Methods
### CFD setup

In this work, a LES of a TBL with a TSB is conducted. Following the LES method, the spatially filtered incompressible non-dimensional Navier–Stokes (NS) equations

$$\nabla \cdot \tilde{\boldsymbol{u}} = 0 \qquad (1)$$

$$\partial_t \tilde{\boldsymbol{u}} + (\tilde{\boldsymbol{u}} \cdot \nabla)\tilde{\boldsymbol{u}} = -\nabla \tilde{p} + Re^{-1}\nabla^2 \tilde{\boldsymbol{u}} - \nabla \cdot \boldsymbol{\tau}, \qquad (2)$$

are solved numerically on a discrete domain. Here, $\tilde{\boldsymbol{u}} = (\tilde{u}, \tilde{v}, \tilde{w})$ is the filtered velocity vector field, $\tilde{p} = \tilde{P}/\rho$ is the density-scaled filtered pressure, and $\boldsymbol{\tau} = \widetilde{\boldsymbol{u} \otimes \boldsymbol{u}} - \tilde{\boldsymbol{u}} \otimes \tilde{\boldsymbol{u}}$ is the sub-grid scale (SGS) stress tensor.

The deviatoric part of the SGS tensor is modeled using the Boussinesq hypothesis

$$\boldsymbol{\tau}^d = \boldsymbol{\tau} - \frac{2}{3}k\boldsymbol{\delta} = -2\nu_{\text{sgs}}\tilde{\boldsymbol{S}}, \tag{3}$$

where $k = \text{tr}(\boldsymbol{\tau})/2$ is the turbulent kinetic energy, $\boldsymbol{\delta}$ is the Kronecker delta, and $\tilde{\boldsymbol{S}} = (\nabla \otimes \tilde{\boldsymbol{u}} + \tilde{\boldsymbol{u}} \otimes \nabla)/2$ is the rate-of-strain tensor. The SGS viscosity is finally closed with the Vreman model[68]. The tilde notation of the LES filtering operation for instantaneous flow-field quantities is thereafter implied.

The high-order spectral-element-method solver SOD2D[57] is used to simulate an incompressible APG TBL in a computational domain with dimensions $L_x \times L_y \times L_z = 1100 \times 120 \times 125$, where $x$ corresponds to the streamwise, $y$ to the wall-normal, and $z$ to the spanwise directions, respectively. A coarse grid and a well-resolved (fine) grid comprising 4th-order hexahedral elements are considered, and details are given in the table of Fig. 7b. The idea of using both a fine and a coarse grid is to perform a cheaper training on the latter which is capable of preserving the main features of the flow so that the trained agent can be effectively applied to the fine-grid case, similar to previous studies[47,60].

The flow solver implements an implicit-explicit Adams–Bransford Crank–Nicolson scheme used together with the fractional-step method to solve the governing equations. In addition, a matrix-free conjugate gradient solver preconditioned by the diagonal is used to solve the linear part of the NS equations and the pressure equation. The computational domain includes a buffer zone of 50 length units in the $x$ direction at the inlet and at the outlet, yielding an effective streamwise domain of 1000 length units ($x \in [-50, 950]$). Dimensions are normalized by the displacement thickness at the inlet $\delta_0^*$, and the velocity field is normalized by the streamwise velocity imposed at the top of the domain, $U_\infty$. The flow is initialized using a Blasius boundary layer at $Re_{\delta_0^*} = 450$, where $Re_{\delta_0^*} = U_\infty \delta_0^*/\nu$ is the Reynolds number based on the displacement thickness. The inlet boundary condition

retains the Blasius profile used as initial condition. At $x = -40$, the laminar boundary layer is tripped using the method explained in refs. 69,70 to accelerate the transition to turbulence. On top the domain, the APG is defined using SB similar to refs. 8,11 as

$$v_{\text{top}} = v_{\text{max}}\sqrt{2}\left(\frac{x_c - x}{\sigma}\right)\exp\left[\psi - \left(\frac{x_c - x}{\sigma}\right)^2\right], \tag{4}$$

where $v_{\text{max}} = 0.4$, $x_c = 306.64$, $\sigma = 110.49$, and $\psi = 0.95$. We note that $v_{\text{max}}$ is 20% larger than the value selected by Wu et al.[11] so that the TSB is still formed when using the coarse LES grid. Moreover, at the top of the domain we apply a zero spanwise vorticity condition ($\omega_z = 0$) and a homogeneous Neumann condition for the spanwise velocity ($\partial_y w = 0$). A periodic boundary condition is imposed in the spanwise direction, and a convective outlet is set on the streamwise end of the domain. On the wall, the lower part of the domain, a classical non-slip boundary condition is imposed. Finally, in the outlet, a pressure-based boundary condition is applied together with the sponge zone. In the fine grid, the a-posteriori computation of the wall-shear stress yields an approximate friction Reynolds number of $Re_\tau = 180$ and $Re_\tau = 750$ at $x = 150$ and $x = 900$, hence right upstream of the TSB and right before the convective outlet of the domain, respectively. A schematic representation of the computational domain which summarizes the setup is shown in Fig. 7, where vortical structures represented with the Q criterion are depicted too.

Once the flow is initialized, the simulation is run until the TSB is formed and the flow has fully developed. Afterwards, the control surfaces start acting on the flow. There are $N_{\text{ac}} = 6$ rectangular actuators located at $x \in [150, 195]$, right upstream of the separation bubble, and each actuator has a spanwise width of $d_{\text{ac}} = 10.42$ which is also the length between adjacent actuators. A classical control strategy similar to those reported in refs. 31,32 is considered as a benchmark. Classical control is implemented as a harmonic time-forcing of the wall-normal velocity, following

$$v_{\text{ac}} = A_{\text{ac}}\sin(2\pi f_{\text{ac}}t), \tag{5}$$

where the free parameters in periodic control are the actuation amplitude $A_{\text{ac}}$ and the actuation frequency $f_{\text{ac}}$. Depending on the APG top-

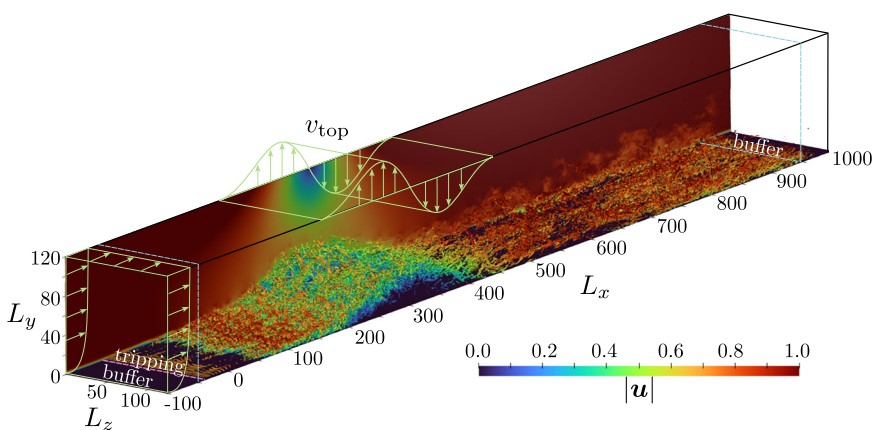

| | $\Delta x^+$ | $\Delta z^+$ | $\Delta y^+\vert_{\text{min}}$ | $\Delta y^+\vert_{\text{max}}$ | DOFs |
|---|---|---|---|---|---|
| Coarse | 38.20 | 38.20 | 1.27 | 25.08 | 4,063,913 |
| Fine | 15.28 | 15.28 | 0.93 | 14.51 | 40,543,349 |

**Fig. 7 | Simulation setup.** Top: schematic representation of the computational domain together with an instantaneous snapshot of vortical structures captured by the Q criterion and colored by velocity magnitude. Bottom: grid details. $\Delta$ refers to grid spacing, the $\cdot^+$ notation refers to viscous units (scaling with $u_\tau$ and $\nu$), and DOFs stands for degrees of freedom, i.e., the total number of nodes in the spectral-element domain discretization.

boundary condition type, the actuation amplitude needs to be adjusted. For example, the SB APG setup requires a larger actuation amplitude because of the forced reattachment arising from the FPG that follows the APG. After a parametric search of an optimal amplitude of actuation, we use $A_{ac} = 0.3U_\infty$, which corresponds to a momentum-flux coefficient of $C_\mu = (v_{ac,rms}/U_\infty)^2 N_{ac}d_{ac}/L_z = 2.25\%$, where 'rms' stands for root-mean-square, which is within the effective range $0.01\% < C_\mu < 3\%$[20,31]. We note that Wu et al.[32] used $C_\mu = 0.0625\%$ in a SO APG setup, and Cho et al.[31] used $C_\mu = 1.4\%$ in a SB APG setup, even though the top wall-normal velocity profile spanned the entire streamwise length of the domain in the latter case. On the other hand, sizing of the control surfaces is based on the objective of effectively modulating the low-frequency motion inherent in the uncontrolled flow, as performed in Wu et al.[32]. These motions are closely associated with the Görtler instability, as reported by Wu et al.[11], which have a spanwise wavelength of the same order of magnitude as the boundary-layer thickness and extends by more than 10 times the boundary-layer thickness in the streamwise direction. More details and comparisons with other control surface implementations can be read in refs. 23,71–75.

In the current setup, we use an actuation frequency of $f_{ac} = 0.0025$ (normalized by $\delta_0^*$ and $U_\infty$). This is the same frequency as identified by Wu et al.[32] for both SO-TSB and SB-TSB configurations (named the "high frequency" in their work). Wu et al.[32] also selected this frequency for the periodic forcing in a SO configuration. In this direction, we perform a spectral analysis of the streamwise velocity component at different locations. While the downstream probe reaches a peak near $f = 0.0015$, a clear dominant frequency cannot be detected. This process is repeated for all the probes displayed in Fig. 3a and the $u$, $v$, $w$, $p$ flow fields, yielding a similar conclusion (not shown here for the sake of brevity). On the other hand, Cho et al.[31] observes that the most effective frequency for the reduction of the TSB in a SB setup is $St = 0.5$. In this work, the non-actuated separation bubble has an approximate length of $L_b = 143$, which translates our selected actuation frequency to $St = 0.3575$, still within the range of effective actuation frequencies proposed by Cho et al.[31].

## DRL setup

Beyond classical periodic forcing, we consider DRL control to reduce the TSB area. As summarized in the introduction, a DRL setup is composed of two main elements: the environment, i.e., the system on which we will apply the control actions, and the agent, i.e., the controller in charge of choosing the actions. In our study, the environment is the simulation performed by the CFD solver, and the agent is a DNN that produces a probability distribution of possible actions. The software used to perform the CFD simulation in the environment is the SOD2D CFD solver[57], and the TF-agents library[58] is used for the DRL model part. A challenge that commonly arises when linking high-performance physics solvers (typically written in Fortran/C/C++) and high-level libraries that implement ML models (typically used through a Python interface) is the communication between the different executable instances, also known as the two-language problem[76]. While this can be accomplished through Unix sockets[53,77] or message-passing interface (MPI)[47], in the current setup the SmartSim[59] library is employed. SmartSim allows communication between processes through an in-memory Redis database with minimal overhead and, compared to the previously mentioned approaches, it lowers the software complexity of the coupling task. RELEXI[51,56] was the first CFD-DRL framework that successfully used SmartSim. SmartSOD2D, which originated as a fork of RELEXI, adapted the framework for the current AFC problem and for the SOD2D CFD solver, hence enabling DRL training on a multi-GPU CFD solver.

In our present implementation, the main computational cost arises from the CFD temporal integration, so the SOD2D solver is run on several GPUs in parallel. On the other hand, the DRL model is based

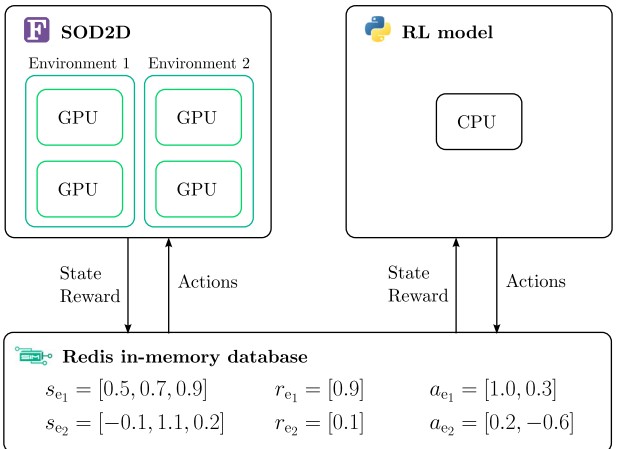

**Fig. 8 | Communication architecture between the CFD environments run in parallel on GPUs and the DRL model run on a CPU.** The in-memory Redis database handles the communication of state, reward, and actions between the solver and the model. For the sake of simplicity, only 2 parallel CFD environments are represented, but we typically run up to 8 parallel CFD environments on the cluster.

on a small multi-layer perceptron DNN and this, together with the reinforcement learning logics around it, can be easily handled by the CPU part of a cluster node which would remain idle otherwise. A schematic diagram of our setup is presented in Fig. 8, noting the multi-environment approach as introduced earlier.

The same actuators as defined for the periodic control are employed in the DRL framework. To impose mass conservation both in space and time, we group the actuators into pairs such that the DRL model sets the action $a$ that represents the mass-flux value on one actuator, and the opposite value ($-a$) is set on the other one. As discussed in the Introduction, the MARL approach can be used when the flow is invariant in a certain direction. In this case, the flow is invariant in the spanwise direction, and a subdomain comprising one pair of actuators is defined as a MARL pseudo-environment. Therefore, a total of $N_{pe} = 3$ MARL pseudo-environments are defined for each CFD environment, hence generating 3 independent trajectories that the agent uses during the optimization step. Also, the output dimensionality is reduced from 3 pairs of jets for the channel as a whole, to 1 pair of jets per MARL pseudo-environment, thus reducing the number of combinations needed to explore all possible control strategies. With the aim of equating the periodic and DRL controls capabilities in terms of energy injected, we allow the agent to select actions within a maximum absolute value equal to the amplitude set in the periodic control, $a = v \in [-v_{ac}, v_{ac}] = [-0.3, 0.3]$. Furthermore, an exponential smoothing function in time is applied to the discrete actions predicted by the agent[37], hence forcing a smooth continuous control signal applied at every time step in the environment.

The environment state $s$ is defined as a set of witness points (probes) that measure the streamwise velocity $u$. The witness points are distributed on a regular grid of size $6 \times 6 \times 6$ points, spanning a $(240 \times 50 \times 104)\delta_0^*$ subdomain. They capture the non-actuated recirculation bubble region and its vicinity, as depicted in Fig. 3a. Each MARL pseudo-environment local state is composed of 2 planes of witness points aligned with the actuators in the streamwise direction, totaling 72 witness points per pseudo-environment.

The reward $r$ is based on the recirculation area of the turbulent bubble. The DRL optimization process aims to maximize $r$ so that the recirculation area is reduced. To calculate a characteristic length for the recirculation bubble ($l_x$), the wall-shear stress value is computed at each element face of the wall surface. For each MARL subdomain, the area of those elements with a negative wall-shear stress $\left(A_i|_{\tau_w < 0}\right)$, i.e., belonging to a local recirculation area, is integrated and then divided

by the MARL subdomain span ($L_{z,pe} = L_z/N_{pe}$)

$$l_x = \frac{1}{L_{z,pe}} \sum_i A_i|_{\tau_w<0}. \quad (6)$$

The reward is normalized by the time-averaged non-actuated characteristic recirculation length $\bar{l}_x^*$, yielding $r = -l_x/\bar{l}_x^*$ (noting that the overline notation indicates a temporal average).

Additionally, instead of using an instantaneous value, the reward is averaged during an actuation period so that the overall response of the system to a given action is more representative. Each MARL pseudo-environment computes a local reward $r_l$ and the global reward $r_g$ is computed as the average of the local rewards. These two reward components, local and global, are combined into the MARL rewards by performing a weighted sum, $r = \alpha r_l + (1 - \alpha)r_g$, where $\alpha$ is a free parameter. We use $\alpha = 0.5$ as this corresponds to an equal importance, for each MARL pseudo-environment, between both the local and global rewards.

An important aspect of the DRL setup is the choice of the policy. A first division is made according to whether the policy is model-based or model-free. While model-based policies rely on a model of the dynamics of the environment, model-free methods optimize the policy through a trial-an-error process in the form of episodes. Model-free policies, such as the one employed in this study, can be further subdivided into policy-gradient, value-function, and actor-critic algorithms, which is a combination of policy-gradient and value-function algorithms. A policy-gradient algorithm is based on the parametrization of the policy and its optimization to maximize cumulative rewards. On the other hand, value-function algorithms try to estimate the cumulative reward given a state (state value function $V^\pi(s)$) or a state-action couple (action value function $Q^\pi(s, a)$). Note that $V^\pi(s)$ and $Q^\pi(s, a)$ are defined as

$$V^\pi(s) = \mathbb{E}_\pi[r|s] \quad (7)$$

$$Q^\pi(s, a) = \mathbb{E}_\pi[r|s, a], \quad (8)$$

where $\mathbb{E}_\pi[\pi]$ denotes the expectation of the reward. An additional subdivision of model-based policies is whether the algorithm is on-policy or off-policy. For on-policy methods, the optimization and policy update are based on the learning generated with that exact same policy. On the other hand, off-policy algorithms use a replay buffer in order to store experiences generated with previous policies which is also used to perform the policy update. Therefore, on-policy algorithms are usually less sample-efficient due to the limited number of experiences used for every policy updated, but they are more stable and simpler to implement. The approach used in this study to optimize the policy is the proximal-policy optimization (PPO)[78]. This is a model-free, actor-critic and on-policy algorithm. The key feature of actor-critic algorithms is that they incorporate two different networks: the actor, which is responsible for interacting with the environment and updating the policy (policy-gradient part), and the critic, which uses a value function, given by the equations (7) or (8), in order to estimate how good the action taken by the actor was (value-function part). Some reasons for choosing PPO over other algorithms like deep deterministic policy gradient (DDPG), twin-delayed DDPG (TD3), or soft actor-critic (SAC), among others, are its small number of tunable parameters, and its suitability for continuous control problems[37,79]. Also, despite being an on-policy algorithm, PPO has been optimized to improve the sample efficiency[78].

We employ a DNN consisting of 2 layers with 128 neurons per layer. An episode duration of $T_e = 4/f_{ac} = 1600$ is used (corresponding to 4 periods of the periodic forcing frequency) so that the low frequencies of the system are captured. The initial condition for each episode is based on a random choice between a fully developed uncontrolled flow

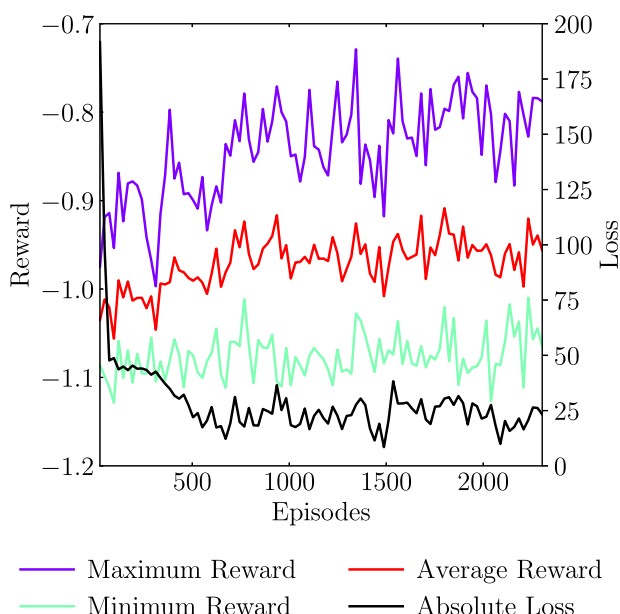

**Fig. 9 | DRL-training metrics.** The average, maximum, and minimum reward signals are extracted from the 24 MARL pseudo-environments running in parallel.

field and the flow field resulting from a previous episode, hence maximizing the exploration space of the model. The actuation frequency of the DRL model is set to $f_{ac,DRL} = 10f_{ac}$, yielding 40 actions per episode, which we considered a trade-off between actuating too often (the flow does not have enough time to develop after a new actuation), and actuating too seldom (the flow is not acted on when required). Actuating every 10% of the system's lowest frequency is within the order of magnitude that can be found in the literature[53].

In Fig. 9, we show the metrics during the training of the DRL model in the coarse grid. We observe that the loss correctly decreases with increasing number of episodes until it stagnates. The average reward metric is a function of the average value of the time-averaged characteristic recirculation lengths provided by the MARL pseudo-environments after an episode, $f(\overline{l_x})$. It is observed that the average reward improves with training (note that, during training, exploration noise is added to the agent, a fact that increases the reward variability) and converges to an average reward lower than $-1$, hence, reducing the TSB characteristic length with respect to the non-actuated baseline case. An improvement is also observed for the maximum reward, while the minimum reward remains approximately stable.

During training, a total of 8 CFD simulations are run in parallel, i.e., $N_e = 8$, hence 24 trajectories are sampled in parallel (since 3 MARL pseudo-environments are used) and a batched optimization step is performed thereafter. This process is repeated 96 times, i.e., a total of 768 CFD episodes are simulated, hence accumulating a total of 2304 MARL episodes. Noting that simulating one episode of single CFD environment takes 1.5 hours on an A100 NVIDIA GPU, the overall training wall time is $1.5 \times 96 = 144$ hours (6 days) when running 8 CFD environments in parallel (one per GPU). This results in a total of 1152 GPU-hours of training.

Regarding the metaparameters of the PPO algorithm, we use the Adam optimizer with a learning rate of $5 \times 10^{-4}$, a discount factor of 0.99, an importance ratio clipping of 0.2, the entropy regularization is set to 0, a lambda value of 0.95 and the rest of the metaparameters are left as default according to the libraries used from TF-Agents[58]. We note that these metaparameter values are typically used in PPO, hence they did not require extensive tuning. More information about these metaparameters can be found in ref. 78 and in the documentation of the libraries in https://github.com/tensorflow/agents.

## Data availability

The data generated in this study have been deposited in the Font et al.[80] database together with the scripts to reproduce the figures in the paper.

## Code availability

The software used in this work is open-source, and the main packages that compose our CFD–DRL framework are listed below. • **SOD2D**: the multi-GPU CFD solver based on the spectral element method. Available in  https://gitlab.com/bsc_sod2d/sod2d_gitlab • **SmartSOD2D**: the communications package for SOD2D based on SmartSim. It allows online training of ML models among other co-processing possibilities. Available in https://github.com/b-fg/SmartSOD2D[81] • **SmartSim**: the workflow library to deploy ML on HPC applications. Available in https://github.com/CrayLabs/SmartSim • **TF-Agents**: the reinforcement learning library based on TensorFlow. Available in https://github.com/tensorflow/agents.

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

## Acknowledgements

This work was performed in part during the Fifth Madrid Summer Workshop, funded by the European Research Council (ERC) under the Caust grant ERC-AdG-101018287. OL has been partially funded by the European Commission's Horizon 2020 Framework program and the

European High-Performance Computing Joint Undertaking (JU) under grant agreement no. 101093393, and by MCIN/AEI/10.13039/501100011033 and the European Union NextGenerationEU/PRTR (PCI2022-134996-2), project CEEC. OL has been partially supported by a Ramon y Cajal postdoctoral contract (ref: RYC2018-025949-I). RV and FAA acknowledge financial support from ERC grant no. '2021-CoG-101043998, DEEPCONTROL'. The authors thank Laia Julió from the BSC support services and Dr. Andrew Shao from the SmartSim team for the help and fruitful discussions about the CFD–DRL software framework. The authors also thank the UPM Turbulence Summer School and our hosts Javier Jiménez, Miguel Pérez Encinar and Adrián Lozano-Durán for the organization of the workshop. Finally, we acknowledge the National Academic Infrastructure for Supercomputing in Sweden (NAISS) for the computational time provided at the large-GPU systems Alvis and Berzelius.

## Author contributions

F.B.: Methodology, framework development, validation, investigation, writing - original draft and visualization. Á-Á.F.: Methodology, framework development, validation, investigation, writing - original draft and visualization. R.J.: Methodology, and writing - review. Vinuesa, R.: Conceptualization, project definition, methodology, resources, project administration, funding acquisition and writing - original draft. L.O.: Methodology, framework development, validation, investigation, resources, writing - original draft, visualization, project administration and funding acquisition.

## Funding

## Competing interests

The authors declare no competing interests.
