## [Transparent Peer Review file · Nature Communications]

Deep reinforcement learning for active flow control in a turbulent separation bubble

Corresponding Author: Dr Bernat Font

Version 0:

Reviewer comments:

Reviewer #1

(Remarks to the Author)

The authors present a pioneering deep reinforcement learning (DRL) study of an important benchmark of flow control, demonstrating significant the superiority of DRL over classical periodic forcing in terms of reduction of the turbulent separation bubble.

To the best of the authors knowledge, this DRL is applied at the highest Reynolds number for this configuration.

The study show the promise of DRL for turbulence control and certainly merits the publication in Nature Communications.

The manuscript is well organized and written.

Publication can be enthusiastically recommended after a minor revision.

Minor comments:

(1) Abstract: Please, elaborate the innovation claims. The Reynolds number is only one of them.

(2) Page 2: The said "climate emergency" is a highly disputed topic as motivation.

The authors are right that one percent saving in fuel leads to 1% saving in government taxes.

(3) Refs [3,4] from 1991, 2011 should be completed by more recent reviews, e.g. Brunton+ 2020 Ann. Rev. Fluid Mech. and Brunton+ 2015 Appl. Mech. Rev.

(4) Page 6: Mentioning DRL as only example of machine learning control does not give adequate credit to recent highly successful developments.

Most experimental feedback turbulence control success stories with many actuators and many sensors are based on other control/ML techniques, either classical control theory for simple dynamics and genetic programming for broad-band frequency turbulence (see comment #3).

(5) Page 8: Why is MARL introduced? MARL is not a novelty claim in the abstract.

(6) Page 10: I would suggest << convective time units (CTU) >>, as the most precise term. There are also viscous time units, etc.

(7) Fig. 1: The captions are hardly readable even with my new glasses. Please enlarge. Please, clearly describe the quantity plotted. $u < 0$ at which height?

(8) Why is the periodic control a good benchmark?

Shouldn't it be feedback phasor control?

Otherwise, the comparison is between open-loop and closed-loop control.

Was the periodic forcing based on optimized parameters?

If yes, which optimized parameters? If no, how can it serve as benchmark?

(9) Fig. 2: What is the significance of the "t" values 12000 left and 2000 right?

The right plots seem to show a transient. When is it converged?

(10) Fig. 3: This "Q" cannot be understood.

One has to read to the end of the paper to get a feel of the meaning.

Please, explain.

(11) Fig. 4: I don't see the need for marking a $k^{-5/3}$ law for a low-Re tripped boundary layer.

There is no constant slope region in the diagram. I propose to remove this law.

(12) Fig. 5: Which significance the the limit 0.3 have?

Please, make figures consistent: Some figures have "t" as lable, this one "Time".

(13) Personally, I don't like to see quantities which are explained much later.

(14) Is the consistent comparison between coarse and fine grids necessary?

If there is a difference, this would make DRL less trustworthy.

If there is no difference, why bother?

(15) Arriving at the end of the paper,

I am still not sure if my interpretation of the solved control problem is right.
I am also missing which mechanisms DRL has learned and
how long it took to do the meta-parameter tuning and final learning.
It does not seem to be phasor control.
Was it an easy or difficult mechanism to be learned?

(Remarks on code availability)

Reviewer #2

(Remarks to the Author)

In their manuscript "Deep reinforcement learning for active flow control in a turbulent separation bubble", Bernat Font et al. demonstrate the efficacy of reinforcement learning (RL) in identifying effective operating conditions for suppressing turbulent flow separation. Optimizing the control parameters of active flow control (AFC) systems has been a research subject for many decades, and it is conceivable that deep learning-based methods can be used to great effect for such optimization tasks. The trial-and-error approach characterizing RL appears to be particularly suited considering recent progress of this method.

The authors apply an RL algorithm (proximal policy optimization) to adjust the mass-flow passing through multiple slots in the surface upstream of a separating and reattaching flow such as to reduce the extent of the latter. The main finding of the study lies in the demonstration that RL is indeed suited to handle this type of optimization, which is reflected in a superior performance of the RL-controlled flow manipulation compared to classical periodic control.

While the subject of the study is certainly of interest to the respective communities, successful demonstrations of RL in the context of AFC have been reported, among others, by Koizumi et al. (2018), Rabault et al. (2019), Fan et al. (2020), Tang et al. (2020) and Amico et al. (2023), the latter of which also addressing a high-Reynolds number scenario. We therefore do not see evidence that the present manuscript advances the current state of the art in a significant way, which is the requirement for publication as a Nature Communication.

Further comments:

1. AFC has been investigated for more than a century, spanning Prandtl's cylinder experiment (1904), Lachmann's seminal compilation (1961), the studies of Wygnanski, Greenblatt and co-workers (late 1990s) and recent studies of controlled turbulent separation bubbles. We feel that this large body of literature is not represented appropriately in the introduction. Specifically, the authors only consider the numerical studies of Wu et al (2020,2022) but do not discuss other studies where AFC is used to control TSBs.

2. In our opinion, the aim to help reduce the aviation carbon footprint with the current effort is commendable, yet very far-fetched:

- i. It is claimed that flow separation occurs on wing surfaces during take-off and landing. The turbulent separation bubble addressed in this study is investigated as a proxy to test control methods that can be applied on aircraft. This raises the question: is flow separation indeed a relevant phenomenon in properly functioning aircraft? A more common motivation to integrate AFC would be to allow for higher angles of attack at a given amount of thrust. In summary, one would strive to prevent flow separation rather than reduce its extent which is done in the present study.
- ii. As reported by the authors, zero-net-mass-flux (ZNMF) devices typically show a periodic mode of operation. There are good reasons for this: first, the reliance on resonance inside the cavity to reach a significant amplitude; second, fluid needs to be ingested in order to be ejected during the subsequent cycle. The authors may want to consider whether a forcing signal with varying amounts of ingested and expelled fluid (figure 5) can be realized in practice or whether ZNMF (per actuator) should be used as a boundary condition during optimization.

3. When reporting the methodology, we feel that several important aspects are missing. First, potential readers may appreciate a brief introduction to the RL algorithm that has been used, including the choices of its main parameters. Second, please explain in detail the action space (also: how was training initiated? How many actions per episode etc.?).

4. The focus point throughout the results section is a comparison between two set-ups of different spatial resolution. In our view, the differences between these two configurations (or lack thereof) should not be treated as the main finding of the study. Instead, it may be reported briefly and serve as a justification to perform RL on coarser-grid simulations.

5. We feel that the visual presentation of results can be improved upon. For example, interesting information could be transmitted in figure 1 by displaying mean and rms values of the wall shear-stress fields rather than iso-surfaces of uniform color for $u < 0$.

(Remarks on code availability)

Reviewer #3

(Remarks to the Author)

(Remarks on code availability)

Reviewer #4

(Remarks to the Author)

- Well written paper, clear and concise
- results are consistent and trustworthy
- w.r.t to fig 5, what is the interpretation / physical reasoning for this control strategy to work?
- The group has significant experience in AFC via RL. I would request to add a short paragraph discussing when to use which RL method, especially PPO vs. TD3 or others?

(Remarks on code availability)

Version 1:

Reviewer comments:

Reviewer #1

(Remarks to the Author)

The authors are commended for their thorough revision with new results addressing all my suggestions. This pioneering work now enjoys an exquisitely polished presentation and is enthusiastically recommended for publication in its present form.

(Remarks on code availability)

Reviewer #2

(Remarks to the Author)

We have carefully read both the revised manuscript and the responses to the referees' comments and are satisfied that the majority of our concerns has been addressed. Yet, we would like to encourage the authors to consider some final minor suggestions as provided below.

1. We understand that the full potential of RL is only leveraged when the forcing signal is not overly constrained. However, we feel that some discussion regarding the implementation of the learnt strategy in real-world applications is required. Please address the following characteristics while doing so:

- The so-called bang-bang actuation (please consider that there is usually some system inertia leading to sinusoidal actuation signals)
- The "instantaneous ZNMF" condition. From our understanding, this boundary condition is required to ensure a stable simulation. However, the operation mode of actual ZNMF actuators would be "ZNMF over one period" because fluid is ingested into a cavity before being reintroduced into the flow.
- The non-periodicity of the actuation signal for each actuator

2. The normalized area of the separation bubble l_x is used as a measure for the control authority. This parameter is called 'length'. Please consider using a different term since what one would consider the TSB length appears to be increased with DRL (figure 1).

3. It is stated on p. 3 that turbulence induces separation (1) and that TSBs occur on wing suction sides (2). Please consider revising as higher turbulence intensity usually increases the near-wall momentum flux making flow separation less likely (1) and reattachment does not typically occur on wings (2), except for laminar separation bubbles.

4. Is the term `_deep_` reinforcement learning really warranted considering the relatively shallow networks used in the present study?

(Remarks on code availability)

N/A

Reviewer #3

(Remarks to the Author)

I co-reviewed this manuscript with one of the reviewers who provided the listed reports. This is part of the Nature

Communications initiative to facilitate training in peer review and to provide appropriate recognition for Early Career Researchers who co-review manuscripts.

(Remarks on code availability)

Reviewer #4

(Remarks to the Author)

I thank the authors for answering my questions and modifying the manuscript. I recommend it for publication.

(Remarks on code availability)

REVIEWER COMMENTS

The authors greatly appreciate the very positive and constructive comments made by all the reviewers. We have addressed each comment individually and modified the manuscript accordingly. Modifications are displayed in **red** for Reviewer #1, **green** for Reviewer #2, and **blue** for Reviewer #4 (Reviewer #3 is omitted due to the shared revision together with another reviewer). Again, thank you very much for taking the time to review this manuscript.

Reviewer #1 (Remarks to the Author):

The authors present a pioneering deep reinforcement learning (DRL) study of an important benchmark of flow control, demonstrating significant the superiority of DRL over classical periodic forcing in terms of reduction of the turbulent separation bubble. To the best of the authors knowledge, this DRL is applied at the highest Reynolds number for this configuration. The study show the promise of DRL for turbulence control and certainly merits the publication in Nature Communications. The manuscript is well organized and written. Publication can be enthusiastically recommended after a minor revision.

Minor comments:

(1) Abstract: Please, elaborate the innovation claims. The Reynolds number is only one of them. The reviewer is correct that the abstract should include all the important findings of the work. In this sense, we have modified it to emphasise the coarse grid (training) / fine grid (testing) strategy, the MARL approach, the physical interpretation of the DRL control, and provided more details of the computational framework.

(2) Page 2: The said "climate emergency" is a highly disputed topic as motivation. The authors are right that one percent saving in fuel leads to 1% saving in government taxes. For the sake of avoiding superficial comments on highly disputed topics, we have removed the sentence. Thanks for pointing it out.

(3) Refs [3,4] from 1991, 2011 should be completed by more recent reviews, e.g. Brunton+ 2020 Ann. Rev. Fluid Mech. and Brunton+ 2015 Appl. Mech. Rev. We thank the reviewer for providing these references. We have added reference Brunton+ 2015 at this point. Reference Brunton+2020 has been added later in the introduction (see comment #4) to support the applicability of machine learning to fluid mechanics and flow control.

(4) Page 6: Mentioning DRL as only example of machine learning control does not give adequate credit to recent highly successful developments. Most experimental feedback turbulence control success stories with many actuators and many sensors are based on other control/ML techniques, either classical control theory for simple dynamics and genetic programming for broad-band frequency turbulence (see comment #3). We agree with the reviewer. The introduction has now been expanded to include a broader overview of ML in FC.

(5) Page 8: Why is MARL introduced? MARL is not a novelty claim in the abstract.

We appreciate this comment by the reviewer. In page 8, we introduce the MARL method since, in our experience, it is of utmost importance when learning AFC strategies for highly stochastic processes in high-dimensional systems. The reviewer is correct that this was not emphasised in the abstract, and while it is not a novelty of this work (as we discuss in page 8), it is definitely an important aspect of it. Hence, we have now acknowledged the usage of MARL in the abstract.

(6) Page 10: I would suggest << convective time units (CTU) >>, as the most precise term.

There are also viscous time units, etc.

This has been changed accordingly, since CTU it is indeed a more accurate term. The change is noted everywhere in the text, and the time axis label in all the relevant figures has been modified as well.

(7) Fig. 1: The captions are hardly readable even with my new glasses. Please enlarge. Please, clearly describe the quantity plotted. $u < 0$ at which height?

The font of the captions, labels, and axis ticks have been enlarged in all figures throughout the manuscript to improve the readability. The first-off wall node height is now specified in the text in p.13. We also note that the respective y^+ value is also given in table inset of Figure 7.

(8) Why is the periodic control a good benchmark? Shouldn't it be feedback phasor control?

Otherwise, the comparison is between open-loop and closed-loop control. Was the periodic forcing based on optimized parameters? If yes, which optimized parameters? If no, how can it serve as benchmark?

We used the paper of Wu et al. 2020 and Wu et al. 2022 as a reference to setup the CFD simulation (domain size and suction and blowing on the top boundary condition) and control (actuator placement and size). In Wu et al. 2022, they use periodic control and, while we do not claim this is the best non-ML control strategy, it can serve as a reference. Indeed, it is an open-loop control strategy, and we compare it with our closed-loop DRL control. Regarding the parameters of the periodic control (amplitude and frequency) we performed a study of which amplitude provided the best results (always in the range of small amplitudes to avoid high injection of momentum). We have added this information in the Methods section and we thank the reviewer for pointing it out. With respect to the frequency we have used, it is the same as in Wu et al. 2022, since that frequency corresponds to the breathing frequency of the bubble (the dominant frequency in the system).

(9) Fig. 2: What is the significance of the "t" values 12000 left and 2000 right? The right plots seem to show a transient. When is it converged?

We are grateful for bringing this point to our attention. Indeed, the original figures did not show the full 20k CTU (originally noted as "t") and this has been fixed in this revision. Furthermore, the labels have been homogenised, and the cases using AFC now include Δ CTU, as an indication that these simulations are restarted from the uncontrolled cases (left plots in Figure 2). The caption of the figure has been amended accordingly. We also note that the previous calculation of the time-averaged characteristic length of the DRL control in the fine grid did not exclude the

initial transient 5k CTU (by accident), and hence this value has been corrected resulting in 9.0% reduction of the TSB compared to the (fine grid) baseline, instead of the 8.9% originally reported. This change has been highlighted in the abstract and noted in the main text as well as in Table 1.

(10) Fig. 3: This "Q" cannot be understood. One has to read to the end of the paper to get a feel of the meaning. Please, explain.

The explanation and reference for the Q criterion has been introduced in Figure 3, when the quantity is first used.

(11) Fig. 4: I don't see the need for marking a $k^{-5/3}$ law for a low-Re tripped boundary layer. There is no constant slope region in the diagram. I propose to remove this law.

Thanks for pointing this out. While we agree that the information conveyed by this line is not significant for this type of flow, we have found in the new figure 5c that the spectra of the DRL control signals closely follow this distribution too. Hence, to indicate such a connection, we have left the $-5/3$ line in both plots and added the corresponding discussion regarding this on p. 22. Also, we note that figure 4 now combines the different resolutions into a single subfigure for each analysed location.

(12) Fig. 5: Which significance the limit 0.3 have? Please, make figures consistent: Some figures have "t" as label, this one "Time".

We are grateful for bringing these points to our attention. The mass flow rate maximum values have been restricted according to the range of momentum flux proposed by Choi et al. 2016, $0.01\% < C_{\mu} < 3\%$. In our case, using $A_{ac}=0.3U$ results in $C_{\mu}=2.25\%$ (hence within this range) and provides the greatest reduction of the TSB in the periodic-control cases. Therefore, the DRL control is also bounded by this same value to allow a fair comparison between both approaches. This information is discussed on p.28-29 within the methods section, but we have now also clarified this in p.19. With respect to the labels, the time label has been fixed throughout the manuscript, now "CTU", and in Fig. 5a we have added the appropriate y axis label.

(13) Personally, I don't like to see quantities which are explained much later. Missing definitions have been now introduced appropriately.

(14) Is the consistent comparison between coarse and fine grids necessary? If there is a difference, this would make DRL less trustworthy. If there is no difference, why bother?

This comparison stems from the following reasoning: we target to control the fine-grid case since it captures a wider range of scales thus being more representative of the physical system. Still, training a DRL-based control agent on the fine grid is not feasible, so we have to use a coarse grid for this. It is important to validate the coarse grid so that the main physics of the problem are still captured, so we include the comparison of both grids for the uncontrolled and control cases. The fact that we do not find major differences at the low-frequency range between both grids justifies the training of the DRL agent on the coarse grid while we target to perform flow control on the fine grid. This reasoning has now been improved in the first paragraph of the Results section.

(15) Arriving at the end of the paper, I am still not sure if my interpretation of the solved control problem is right. I am also missing which mechanisms DRL has learned and how long it took to do the meta-parameter tuning and final learning. It does not seem to be phasor control.

Was it an easy or difficult mechanism to be learned?

We agree that the physical interpretation of the DRL control was missing in the original manuscript and we have now added an in-depth analysis of the control signals and DRL control flow fields to address this. Please see the new figures 5b,c and 6, and the interpretation of these results in p. 20-23. In short, we find that the actuator pairs form streamwise vortices that interact with the TSB. The control signal exhibits local peaks near the natural frequency of the TSB (fig. 5c) while trying to sustain these structures for long time periods (fig 5b). This is then visualised by the average vorticity contour in fig. 6. These figures are included below for reference. Regarding the DRL meta-parameter tuning and training cost, this has now been detailed in the “Methods - DRL setup” section.

“Figure 5. Analysis of actuation signals. (a) Temporal signal of the DRL-based actuators for the fine-grid (left) and coarse-grid (right) cases. The exponential smoothing between two discrete actions that sets the instantaneous mass-flow rate of the actuators is also displayed. Note that actuators 1 and 3 overlap on the fine-grid case. (b) Normalized cross-correlation between actuation signals of the coarse-grid case, where ξ is the correlation interval. (c) Power spectrum of the actuation signals of the coarse-grid case computed with the Welch method using 6 splits with 50% overlap and no windowing.”

Figure 6. Time-averaged coarse-grid flow fields for the DRL control (top) and uncontrolled (bottom). The x-axis is aligned with the streamwise flow direction. The first and second columns show the time-averaged streamwise and wall-normal velocity components, respectively. The third column shows the iso-surface of the time-averaged spanwise velocity $|w|=0.02$ (red for positive), while the wall and the periodic plane are colored by u . The fourth column shows the iso-surface of the time-averaged vorticity magnitude $|\omega|=0.04$, while the wall and the periodic plane are colored by u .

REVIEWER COMMENTS

The authors greatly appreciate the very positive and constructive comments made by all the reviewers. We have addressed each comment individually and modified the manuscript accordingly. Modifications are displayed in **red** for Reviewer #1, **green** for Reviewer #2, and **blue** for Reviewer #4 (Reviewer #3 is omitted due to the shared revision together with another reviewer). Again, thank you very much for taking the time to review this manuscript.

Reviewer #2 (Remarks to the Author):

In their manuscript “Deep reinforcement learning for active flow control in a turbulent separation bubble“, Bernat Font et al. demonstrate the efficacy of reinforcement learning (RL) in identifying effective operating conditions for suppressing turbulent flow separation. Optimizing the control parameters of active flow control (AFC) systems has been a research subject for many decades, and it is conceivable that deep learning-based methods can be used to great effect for such optimization tasks. The trial-and-error approach characterizing RL appears to be particularly suited considering recent progress of this method.

The authors apply an RL algorithm (proximal policy optimization) to adjust the mass-flow passing through multiple slots in the surface upstream of a separating and reattaching flow such as to reduce the extent of the latter. The main finding of the study lies in the demonstration that RL is indeed suited to handle this type of optimization, which is reflected in a superior performance of the RL-controlled flow manipulation compared to classical periodic control.

While the subject of the study is certainly of interest to the respective communities, successful demonstrations of RL in the context of AFC have been reported, among others, by Koizumi et al. (2018), Rabault et al. (2019), Fan et al. (2020), Tang et al. (2020) and Amico et al. (2023), the latter of which also addressing a high-Reynolds number scenario. We therefore do not see evidence that the present manuscript advances the current state of the art in a significant way, which is the requirement for publication as a Nature Communication

We thank the reviewer for bringing this important point into our attention. The cited publications have indeed demonstrated the success of RL control for complex flows. Still, in the present work, we demonstrate the RL control of a highly turbulent flow using a purely numerical approach, in contrast to the work of Amico et al. which uses RL control in an experimental setup. This numerical demonstration, for the first time applied at such a high Reynolds number, sets a precedent on the feasibility of using RL control for the early stage of a design cycle, when the cost of experimental setups can often limit the viability of a project. We note that this has been possible as a result of combining a very fast multi-GPU CFD solver, a parallel multi-environment strategy, and the multi-agent RL approach. In our opinion, this framework serves as a prime example of the efficacy of RL control for scale-resolving fluid-flow simulations, significantly advancing the state of the art. On the other hand, we agree that the literature cited in the original manuscript was not representative enough, and, additionally to Rabault et al. (2019) and Tang et al. (2020) which were already considered in the original manuscript, the works of Koizumi et al. (2018), Fan et al. (2020), and Amico et al. (2023), have now been included in the literature review.

Further comments:

1. AFC has been investigated for more than a century, spanning Prandtl's cylinder experiment (1904), Lachmann's seminal compilation (1961), the studies of Wygnanski, Greenblatt and co-workers (late 1990s) and recent studies of controlled turbulent separation bubbles. We feel that this large body of literature is not represented appropriately in the introduction. Specifically, the authors only consider the numerical studies of Wu et al (2020,2022) but do not discuss other studies where AFC is used to control TSBs.

We thank the reviewer for pointing this out. We have added a discussion about AFC for separated flows with more references, also introducing some of the different methods in the literature and giving a broader overview on the ZNMF periodic control.

2. In our opinion, the aim to help reduce the aviation carbon footprint with the current effort is commendable, yet very far-fetched:

We agree with this take, and for the sake of avoiding superficial comments on highly disputed topics, we have removed the sentence "Furthermore, it is even more crucial to tackle this problem in the context of the current climate emergency."

i. It is claimed that flow separation occurs on wing surfaces during take-off and landing. The turbulent separation bubble addressed in this study is investigated as a proxy to test control methods that can be applied on aircraft. This raises the question: is flow separation indeed a relevant phenomenon in properly functioning aircraft? A more common motivation to integrate AFC would be to allow for higher angles of attack at a given amount of thrust. In summary, one would strive to prevent flow separation rather than reduce its extent which is done in the present study.

We agree that preventing flow separation would always be more beneficial rather than having to reduce it. But we also think that not in all cases it is possible to achieve it. As the literature about separation-control reflects, this is a phenomenon intrinsic to many industrial applications and therefore, we believe it is important to tackle it. We have added this to the article and, as suggested in the previous point, we have added a small discussion about AFC for separated flows which help to support the use of the control in a separated flow.

ii. As reported by the authors, zero-net-mass-flux (ZNMF) devices typically show a periodic mode of operation. There are good reasons for this: first, the reliance on resonance inside the cavity to reach a significant amplitude; second, fluid needs to be ingested in order to be ejected during the subsequent cycle. The authors may want to consider whether a forcing signal with varying amounts of ingested and expelled fluid (figure 5) can be realized in practice or whether ZNMF (per actuator) should be used as a boundary condition during optimization.

We thank the reviewer for pointing this out. Regarding the practicality of the action signal, we note that the DRL control actuators are paired so that the action imposed in one of them is the opposite action imposed in the control surface pair. This makes the DRL control to be ZNMF

instantaneously. As we mention in comment#3, we have clarified this in the methodology so it is more understandable.

3. When reporting the methodology, we feel that several important aspects are missing. First, potential readers may appreciate a brief introduction to the RL algorithm that has been used, including the choices of its main parameters. Second, please explain in detail the action space (also: how was training initiated? How many actions per episode etc.?).

We appreciate the comment of the reviewer and we think that including these aspects is necessary and will improve the quality of the manuscript. We have added a discussion about the policies and why we choose PPO in the Methods section. Also, we have added a last paragraph in the methodology with the values of the metaparameters used for our PPO algorithm. The action space of each agent is of dimension 1 (since we use MARL) and the range of actions is $[-0.3, 0.3]$, same as the periodic-control amplitude. We have modified the paragraph about it and we think that it is clearer now. To initialize each training episode, a random choice between a fully developed uncontrolled flow field and the flow field resulting from a previous DRL-controlled episode is made, hence maximizing exploration space of the DRL model. Regarding the number of actions per episode, we use a total of 40, which we considered as a trade-off between actuating too often (the flow does not have enough time to develop after a new actuation), and actuating too seldom (the flow is not acted on when required). This is noted in the DRL methodology as well.

4. The focus point throughout the results section is a comparison between two set-ups of different spatial resolution. In our view, the differences between these two configurations (or lack thereof) should not be treated as the main finding of the study. Instead, it may be reported briefly and serve as a justification to perform RL on coarser-grid simulations.

We agree with the reviewer that the main point of the results is not to compare two different grid resolutions. In this sense, we have improved our explanation on the use of two different grids. The improved explanation can be found in the first paragraph of the results section. Overall, this should give a clearer dissection of our results, which indeed must focus on the successful application of DRL control for the TSB.

5. We feel that the visual presentation of results can be improved upon. For example, interesting information could be transmitted in figure 1 by displaying mean and rms values of the wall shear-stress fields rather than iso-surfaces of uniform color for $u < 0$.

The reviewer is correct that flow visualization could be improved. For this, we have added the new figure 6 (included below for reference) which, together with figures 3, and 9, helps to better understand the flow structures for all the three different cases: uncontrolled, periodic control, and DRL control. In addition, this work has now allowed us to improve our physical interpretation of the DRL control mechanism, which was a missing part of the manuscript as also pointed out by reviewers 1 and 3. Hence, we would like to thank the reviewer for pushing us to improve the visual presentation of the results.

Figure 6. Time-averaged coarse-grid flow fields for the DRL control (top) and uncontrolled (bottom). The x-axis is aligned with the streamwise flow direction. The first and second columns show the time-averaged streamwise and wall-normal velocity components, respectively. The third column shows the iso-surface of the time-averaged spanwise velocity $|w|=0.02$ (red for positive), while the wall and the periodic plane are colored by u . The fourth column shows the iso-surface of the time-averaged vorticity magnitude $|\omega|=0.04$, while the wall and the periodic plane are colored by u .

REVIEWER COMMENTS

The authors greatly appreciate the very positive and constructive comments made by all the reviewers. We have addressed each comment individually and modified the manuscript accordingly. Modifications are displayed in **red** for Reviewer #1, **green** for Reviewer #2, and **blue** for Reviewer #4 (Reviewer #3 is omitted due to the shared revision together with another reviewer). Again, thank you very much for taking the time to review this manuscript.

Reviewer #3 (Remarks to the Author):

I co-reviewed this manuscript with one of the reviewers who provided the listed reports. This is part of the Nature Communications initiative to facilitate training in peer review and to provide appropriate recognition for Early Career Researchers who co-review manuscripts. We thank the Reviewer for their assessment of our work, and we hope that we could address all the comments to their satisfaction.

REVIEWER COMMENTS

The authors greatly appreciate the very positive and constructive comments made by all the reviewers. We have addressed each comment individually and modified the manuscript accordingly. Modifications are displayed in **red** for Reviewer #1, **green** for Reviewer #2, and **blue** for Reviewer #4 (Reviewer #3 is omitted due to the shared revision together with another reviewer). Again, thank you very much for taking the time to review this manuscript.

Reviewer #4 (Remarks to the Author):

- Well written paper, clear and concise
- results are consistent and trustworthy

We appreciate the kind and positive comments by the reviewer.

- w.r.t to fig 5, what is the interpretation / physical reasoning for this control strategy to work? The physical interpretation of the DRL control strategy has now been expanded based on the new figures 5 and 6 (included below for reference), and the accompanying discussion on p.20-23. We thank the reviewer for bringing this important point into our attention.

“Figure 5. Analysis of actuation signals. (a) Temporal signal of the DRL-based actuators for the fine-grid (left) and coarse-grid (right) cases. The exponential smoothing between two discrete actions that sets the instantaneous mass-flow rate of the actuators is also displayed. Note that actuators 1 and 3 overlap on the fine-grid case. (b) Normalized cross-correlation between

actuation signals of the coarse-grid case, where ξ is the correlation interval. (c) Power spectrum of the actuation signals of the coarse-grid case computed with the Welch method using 6 splits with 50% overlap and no windowing.”

Figure 6. Time-averaged coarse-grid flow fields for the DRL control (top) and uncontrolled (bottom). The x-axis is aligned with the streamwise flow direction. The first and second columns show the time-averaged streamwise and wall-normal velocity components, respectively. The third column shows the iso-surface of the time-averaged spanwise velocity $|w|=0.02$ (red for positive), while the wall and the periodic plane are colored by u . The fourth column shows the iso-surface of the time-averaged vorticity magnitude $|\omega|=0.04$, while the wall and the periodic plane are colored by u .

- The group has significant experience in AFC via RL. I would request to add a short paragraph discussing when to use which RL method, especially PPO vs. TD3 or others?

We agree that more information on the RL method can help the readers to better understand the motivation behind the chosen method. In this sense, we have added a discussion about the policies and a brief comparison with a justification of why we use PPO in the Methods.

REVIEWER COMMENTS

The authors greatly appreciate once again the reviewers for their work, and we are happy to read that the revised manuscript has been generally recommended for publication. We have addressed the remaining minor comments from Reviewer #2 in green. Again, thank you very much for taking the time to review this manuscript.

Reviewer #2 (Remarks to the Author):

We have carefully read both the revised manuscript and the responses to the referees' comments and are satisfied that the majority of our concerns has been addressed. Yet, we would like to encourage the authors to consider some final minor suggestions as provided below.

1. We understand that the full potential of RL is only leveraged when the forcing signal is not overly constrained. However, we feel that some discussion regarding the implementation of the learnt strategy in real-world applications is required. Please address the following characteristics while doing so:

We agree with the reviewer that some additional comments on the implementation of RL control in real-world applications can improve the manuscript. For this, we have added additional information in the introduction commenting on the challenges related to experimental DRL setups (and added two references for this).

a. The so-called bang-bang actuation (please consider that there is usually some system inertia leading to sinusoidal actuation signals)

We thank the reviewer for bringing this point to our attention. We note that the instantaneous DRL control is smoothed using an exponential function, as described in p.33. Hence the system inertia is, to some extent, present in the current control function. For clarification, we have added this on the paper when discussing the bang-bang control:

“Note that despite the actuation signal resembling a bang–bang control, the actual values imposed on the control surfaces are smoothed over time, as explained later in the Methods section. This results in a more realistic and applicable control signal.”

b. The “instantaneous ZNMF” condition. From our understanding, this boundary condition is required to ensure a stable simulation. However, the operation mode of actual ZNMF actuators would be “ZNMF over one period” because fluid is ingested into a cavity before being reintroduced into the flow.

The reviewer is correct that the ZNMF is acting over a period, and we mention this explicitly in p.5 and p.18. To avoid confusion and reduce the acronyms in the abstract, we have removed ZNMF from the abstract and clarified this with the following sentence: “Furthermore, the DRL agent provides a smoother control strategy while conserving momentum instantaneously.”

c. The non-periodicity of the actuation signal for each actuator

We thank the reviewer for pointing this out. We agree that a non-periodic temporal actuation signal is always going to be more complex in a real world application. We have added a short discussion in the introduction and references to other works that comment about this including the framework presented by Dong et. al., PoF 2024 “An interactive platform of deep reinforcement learning and wind tunnel testing”.

2. The normalized area of the separation bubble I_x is used as a measure for the control authority. This parameter is called ‘length’. Please consider using a different term since what one would consider the TSB length appears to be increased with DRL (figure 1).

The authors thank the reviewer for spotting this. The full description for I_x is actually “characteristic recirculation length”, where ‘characteristic’ denotes a scale quantity rather than a physically measurable length. We also include ‘length’ in its description since its units are $[L]$ and not $[L^2]$. This is clarified in p.12 when first introduced. In addition, we have changed “TSB length” to “TSB area” when appropriate throughout the manuscript.

3. It is stated on p. 3 that turbulence induces separation (1) and that TSBs occur on wing suction sides (2). Please consider revising as higher turbulence intensity usually increases the near-wall momentum flux making flow separation less likely (1) and reattachment does not typically occur on wings (2), except for laminar separation bubbles.

The reviewer is correct that the current phrasing could be misinterpreted thus yielding the described inaccuracy. For this, we have changed “...or due to turbulence.” to “...or due to large-scale turbulence in the free-stream flow”.

4. Is the term deep reinforcement learning really warranted considering the relatively shallow networks used in the present study?

We thank the reviewer for bringing up this discussion. We understand that using the term “deep” in our case of 2 hidden layers does not compare well with larger models, which can use $O(10-10^3)$. Still, all artificial neural networks with more than 1 hidden layer are commonly described as “deep” since their non-linear nature is remarkably different from the single hidden layer networks. Hence, we have opted for keeping the term “deep” in the manuscript, which is also consistent with our previous publications in flow control.